# Droplet-based high-throughput cultivation for accurate screening of antibiotic resistant gut microbes

William J Watterson[1,2]\*, Melikhan Tanyeri[1,2,3], Andrea R Watson[4],
Candace M Cham[4], Yue Shan[4], Eugene B Chang[4], A Murat Eren[4,5,6]\*, Savaş Tay[1,2]\*

[1]Pritzker School of Molecular Engineering, The University of Chicago, Chicago, United States; [2]Institute for Genomics and Systems Biology, The University of Chicago, Chicago, United States; [3]Department of Engineering, Duquesne University, Pittsburgh, United States; [4]Department of Medicine, The University of Chicago, Chicago, United States; [5]Graduate Program in the Biophysical Sciences, The University of Chicago, Chicago, United States; [6]Josephine Bay Paul Center for Comparative Molecular Biology and Evolution, Marine Biological Laboratory, Woods Hole, United States

\*For correspondence:
william.j.watterson@gmail.com (WJW);
meren@uchicago.edu (AME);
tays@uchicago.edu (ST)

**Abstract** Traditional cultivation approaches in microbiology are labor-intensive, low-throughput, and yield biased sampling of environmental microbes due to ecological and evolutionary factors. New strategies are needed for ample representation of rare taxa and slow-growers that are often outcompeted by fast-growers in cultivation experiments. Here we describe a microfluidic platform that anaerobically isolates and cultivates microbial cells in millions of picoliter droplets and automatically sorts them based on colony density to enhance slow-growing organisms. We applied our strategy to a fecal microbiota transplant (FMT) donor stool using multiple growth media, and found significant increase in taxonomic richness and larger representation of rare and clinically relevant taxa among droplet-grown cells compared to conventional plates. Furthermore, screening the FMT donor stool for antibiotic resistance revealed 21 populations that evaded detection in plate-based assessment of antibiotic resistance. Our method improves cultivation-based surveys of diverse microbiomes to gain deeper insights into microbial functioning and lifestyles.

## Introduction

Culture-independent surveys of naturally occurring microbial populations through marker gene amplicons and shotgun metagenomes have revealed intriguing associations between the gut microbial communities and human health (*Knight et al., 2017*; *Lynch and Pedersen, 2016*). However, inferring the taxonomic composition or functional potential of complex gut microbiomes does not reveal mechanistic underpinnings of observed associations (*Surana and Kasper, 2017*; *Ni et al., 2017*; *Schmidt et al., 2018*). One of the essential steps to address such shortcomings is microbial cultivation, which enables the recovery of complete reference genomes (*Mukherjee et al., 2017*), accurate identification of taxonomy and functional potential of new strains (*Forster et al., 2019*; *Zou et al., 2019*), and validation of causality through perturbation experiments (*Schmidt et al., 2018*). Microbial cultivation is currently experiencing a pronounced revival (*Forster et al., 2019*; *Zou et al., 2019*; *Browne et al., 2016*; *Lagier et al., 2016*; *Villa et al., 2020*), yet the majority of cultivation efforts that rely on traditional cultivation strategies require arduous manual picking of thousands of colonies, impeding the efforts to harmonize discoveries that emerge from 'omics strategies with downstream mechanistic investigations in the rapidly advancing field of microbiome research.

**eLife digest** The human gut is inhabited with hundreds of billions of bacterial cells from a wide range of families. This complex mixture of bacteria is part of the gut microbiome, along with other lifeforms such as viruses, archaea and fungi. As well as interacting with each other, the bacteria in the microbiome interact with our cells and available nutrients. Studying these interactions can help us understand how this community of bacteria influence health and disease.

One way to study the diversity of the microbiome is to take a sample, such as a section of stool, and perform DNA sequencing to determine which types of bacteria are present. This can reveal how the composition of the gut microbiome relates to our health, but cannot confirm whether these bacteria are the cause or the effect of most diseases.

To overcome this problem, researchers need to be able to grow pure strains of these bacteria in order to unravel their underlying mechanisms. For over a century, the conventional way to cultivate bacteria has been to grow them in a Petri dish. However, this method promotes the growth of more abundant, fast-growing bacterial strains. This results in a huge disconnect between the bacteria grown in a Petri dish and the diversity within the human gut, which is hindering our understanding of gut health and disease.

Now, Watterson et al. have built a machine that improves the speed and number of cultivated bacterial organisms, thus paving the way for more detailed investigations of the human gut microbiome. This new system works by growing bacteria in millions of miniscule droplets which can be physically separated to help the expansion of slower growing species.

Watterson et al. cultivated bacterial cells from a stool sample from a single donor using the droplet system and compared this to traditional culturing methods. The droplet technology increased the number of different organisms that were able to grow by up to four times, including those that were rare or slow-growing. Bacteria in the donor stool were then screened for populations that were resistant to antibiotics. This identified 21 antibiotic resistant bacteria which only grew in the droplets and not in Petri dishes.

This droplet-based technology will make it possible to study bacterial strains that were previously difficult to grow. Furthermore, this method could help identify whether stool from a donor contains any antibiotic resistant strains, which can lead to clinical complications once transplanted. In future, this new technology could be used in laboratories or hospitals to study the role of the gut microbiome in health and disease.

Recent years have witnessed numerous new cultivation strategies that increase the throughput in isolating and studying gut-associated bacteria. For example, a recent well plate-based growth experiment screened 96 phylogenetically diverse human gut-associated bacterial strains across 19 media and determined their nutritional preferences and biosynthetic capabilities (*Tramontano et al., 2018*). Another study that relied on 'SlipChip' (*Du et al., 2009*), a microfluidic device that can isolate hundreds of microbial cells and enable targeted cultivation, successfully recovered an organism that was a member of the genus *Oscillibacter* (*Ma et al., 2014*), which had been one of the 'most wanted taxa' from the human gut (*Fodor et al., 2012*), a list of uncultivated yet highly prevalent taxa from the Human Microbiome Project (*Huttenhower et al., 2012*). Biomimetic devices represent another active area of research (*Bhatia and Ingber, 2014*). For instance, the 'gut-on-a-chip' offers a controlled microfluidics platform which mimics the physical and functional features of the intestinal environment and enables complex in vitro chemical gradients and multicellular interactions (*Kim et al., 2012*; *Kim et al., 2016*) that can establish stable co-culturing of complex bacterial populations (*Jalili-Firoozinezhad et al., 2019*). Although these techniques increase the throughput in isolating and manipulating gut organisms as compared to plate-based culture, their throughput is insufficient for isolating rare organisms among the thousands of gut-associated species or performing large-scale perturbation experiments.

Droplet microfluidics offers a promising alternative for high-throughput anaerobic cultivation. The aqueous droplets, with typical volumes ranging from picoliters to nanoliters, are generated and manipulated with an oil phase in microfluidic channels. An extensive arsenal of droplet microfluidic tools has been developed for use in standard aerobic lab environments, where oxygen is present

(*Kaminski et al., 2016*). For instance, droplets can be generated and sorted at rates exceeding 10 kHz (*Sciambi and Abate, 2015*), reagents can be added by pico-injection or droplet merging (*Abate et al., 2010*; *Niu et al., 2008*), and the droplets can further be stored in a regular array and retrieved for downstream applications (*Jiang et al., 2016*; *Cole et al., 2017*). Droplets also eliminate a major bottleneck of conventional broth and plate-based culture: the overgrowth of fast-growing populations over slow-growers. In particular, the stochastic isolation of individual bacterial cells in discrete droplets prior to cultivation eliminates the competition that favors fast-growers and yields more accurate representation of the distribution of microbial cells from the input sample (*Jiang et al., 2016*; *Zengler et al., 2002*). For instance, *Jiang et al., 2016* isolated environmental soil-associated bacteria in droplets and found an increase in the diversity of taxa with an increased representation of rare organisms (*Jiang et al., 2016*). *Villa et al., 2020* recently cultivated human gut microbes in thousands of nanoliter droplets to characterize metabolic variation in polysaccharide-degrading gut bacteria and analyzed their growth kinetics (*Villa et al., 2020*). Due to small droplet volume, bacteria confined within droplets can reach a critical threshold concentration of quorum sensing molecules faster than would occur in bulk culture, which can lead to improved growth in certain culture medium (*Boedicker et al., 2009*). These key advantages afforded by droplet microfluidics thus present an ideal technology to improve the speed and efficiency of traditional strategies used for anaerobic cultivation.

Here, we present an end-to-end platform for high-throughput automated isolation, cultivation, and sorting of anaerobic bacteria in microfluidic droplets. The technology is comprised of droplet microfluidic devices operated inside of an anaerobic chamber and an automated rapid image processing system. We characterized our technology's ability using a stool sample from a human subject and three different growth media. The droplet-based anaerobic isolation (i) achieved a larger representation of microbes in the original stool sample compared to traditional cultivation strategies regardless of the growth medium, (ii) promoted the growth of a larger fraction of rare and slow-growing taxa in the original sample, and (iii) detected significantly more antibiotic resistant strains from the stool sample than could be detected through traditional plate-based cultivation. Overall, droplet-based cultivation has the potential to increase the throughput and accuracy of cultivating pure strains from anaerobic environments. As fecal microbiome transplant is becoming an increasingly powerful approach for the treatment of several gut conditions such as *Clostridium difficile* colitis, there is great need for rapidly and affordably screening of these complex microbial populations. Our technology enables rapid and efficient screening for antibiotic resistant microbes in donor stool samples and improves the safety of fecal microbial transplant treatments.

## Results

### Isolation, culture, and sorting of anaerobes in a high-throughput droplet microfluidic system

We developed an array of droplet microfluidic technologies for the high-throughput cultivation and manipulation of anaerobic microbial communities (*Figure 1*). The microfluidic devices are housed within an anaerobic chamber along with a microscope, syringe pumps, a high frame rate camera, electrodes, and an incubator (*Figure 1a*). A computer external to the anaerobic chamber controls the camera, syringe pumps, and electrodes (via a voltage amplifier also external to the chamber). We generated droplets at a flow focusing junction from liquid culture medium into oil (*Figure 1b*). Our strategy stochastically encapsulated microbial cells in the droplets (~65–115 pL) by diluting fecal cell suspensions or liquid broth cultures in the medium so that ~2% to 12% of droplets initially contain only one live bacterial cell according to Poisson statistics. We then placed the droplet emulsion in an incubator at 37°C and the isolated viable strains can clonally replicate within a droplet provided the strain can grow within the environmental conditions (*Figure 1c*). Since each colony is isolated within a confined droplet, the slow-growing populations avoid the competitive overgrowth of fast-growing populations – which often occurs in traditional broth or Petri agar cultivation. We validated the anaerobic isolation and culture method using extremely oxygen sensitive anaerobes from the model mouse gut microbiota community, the Altered Schaedler's Flora (*Wymore Brand et al., 2015*; *Figure 1—figure supplement 1*). The droplets remained stable for several days in culture,

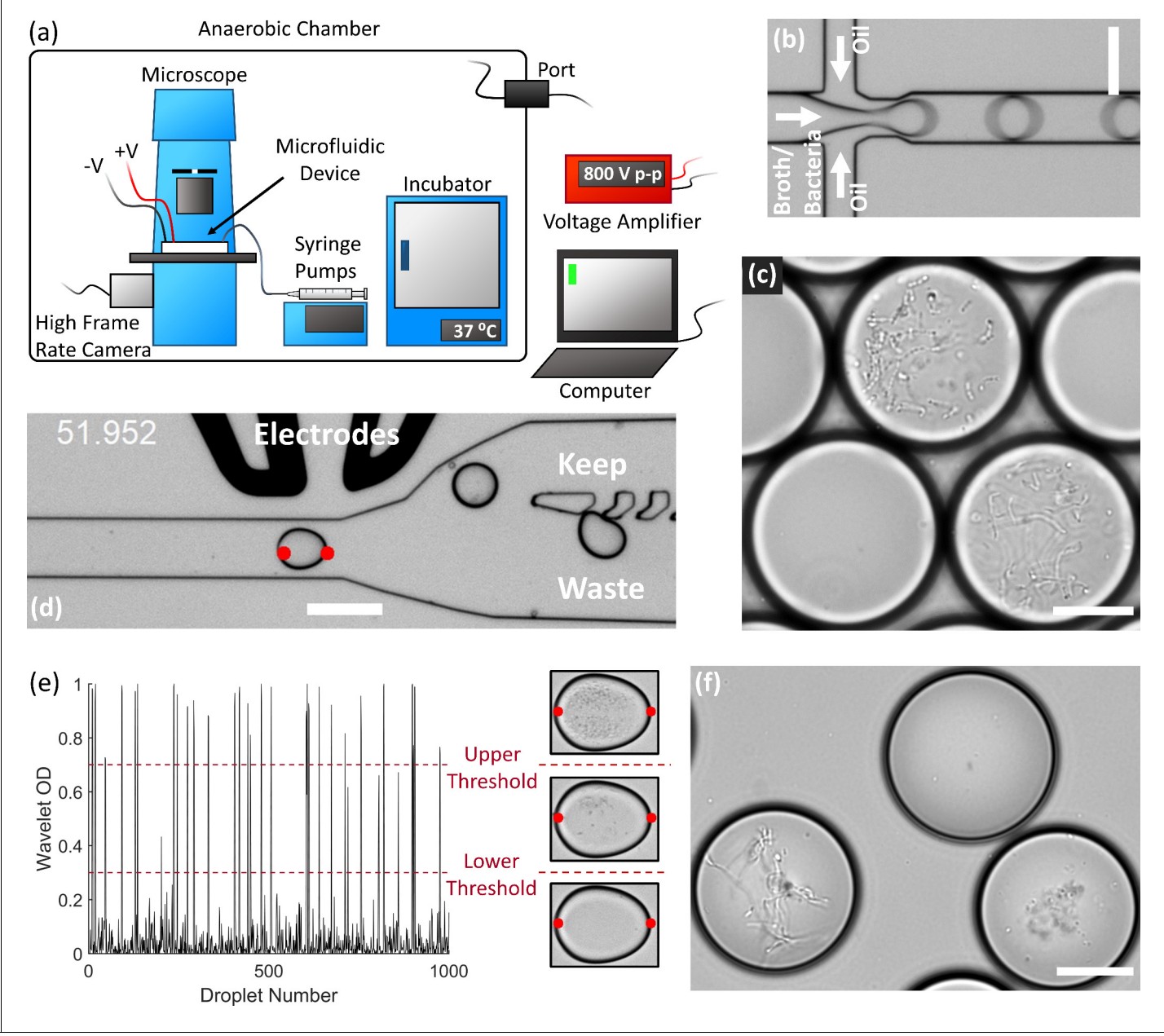

**Figure 1.** End-to-end system for efficient isolation and culture of gut anaerobes in microfluidic droplets. (a) The experimental setup for isolating, culturing, and sorting anaerobic bacteria in microfluidic droplets consists of a microscope, microfluidic devices, a high frame rate camera, syringe pumps, an incubator, and electrodes all contained within an anaerobic chamber. The computer controls the syringe pumps, high frame rate camera, and electrodes (via a voltage amplifier). The equipment power and control wires are introduced to the anaerobic chamber through sealed rubber ports to strictly maintain the anaerobic conditions within the chamber. (b) Single bacteria cells are isolated in droplets containing anaerobic culture medium and the resulting emulsion is cultured inside the incubator. (c) An example of human gut bacteria isolated and cultivated inside droplets. (d) Droplets are sorted by optical detection and subsequent deflection via dielectrophoresis near a sorting junction. Specifically, droplets with bacterial colonies which meet a certain thresholding criteria were determined using image analysis (region between the red dots), and these droplets were deflected into the 'keep' path by actuating an on-chip electrode while sending the remaining droplets to waste. (e) The colony density measured by image analysis (Wavelet OD) for 1000 successive droplets. Droplets with a dense colony, a sparse colony, and empty droplets (no colony) are represented by a wavelet OD value above an upper threshold, between an upper and lower threshold, or below a lower threshold, respectively. (f) Two slow-growing human gut-associated bacteria colonies (bottom left and bottom right) grown in droplets after sorting and a false positive empty droplet (top). Scale bar in (b) and (d) is 100 µm and scale bar in (c) and (f) is 20 µm.

The online version of this article includes the following figure supplement(s) for figure 1:

*Figure 1 continued on next page*

Figure 1 continued
**Figure supplement 1.** The Altered Schaedler's Flora, ASF, is an important and widely studied gnotobiotic mouse model used for understanding microbiota-host dynamics in both health and disease.
**Figure supplement 2.** Droplet volume decreased during cultivation in the anaerobic chamber.

although by 4 days the arid atmosphere of the anaerobic chamber led to a significant reduction in droplet volume due to evaporation (*Figure 1—figure supplement 2*).

We also developed an image-based sorting algorithm and microfluidic control system for sorting bacterial colonies in droplets based on the colony density (*Figure 1d*). Importantly, image-based droplet sorting does not require fluorescent strains or reporters and therefore has broad applicability in processing environmental samples (*Zang et al., 2013*). The high-frame rate camera along with a custom LabView code automatically detects the droplet approaching the sorting junction and performs a wavelet-based image analysis of an optical density-like measurement, which we termed the Wavelet OD (see Materials and methods). If the Wavelet OD satisfies an empirically-defined thresholding criteria, the computer will actuate the electrodes via a voltage amplifier and deflect the droplet bacteria colony into the 'keep' path (*Figure 1d*). The Wavelet OD value varies between 0 (empty droplets) and 1 (droplets with a very dense colony). Droplets were sorted at a rate of ~30 Hz. We retrieved slow-growing colonies by sorting droplets with a Wavelet OD within an empirically defined lower and upper threshold value (*Figure 1e–f*).

## Cultivation of human stool microbiota in droplets enhances richness and abundance of rare taxa

To explore the growth potential of human gut bacteria in microfluidic droplets, we used a single human fecal sample from a previously characterized FMT donor (*Lee et al., 2017*) and cultivated microorganisms in droplets composed of three rich media up to 2 days (*Figure 1c*, *Figure 2—figure supplement 1*, and *Video 1*). In parallel, we used the same set of media on plates for cultivation from the same sample for up to seven days. Our three rich media included Brain Heart Infusion Supplemented (BHIS), Gut Microbiota Medium (GMM), and Yeast Casitones Fatty Acid (YCFA). After cultivation, we extracted the genomic DNA from plate scrapings or by breaking the pooled droplet emulsion. The samples were sequenced using paired-end 16S rRNA gene sequencing on the Illumina platform with primers targeting the V4 region. To infer highly resolved microbial community structures in our amplicon data, we used Minimum Entropy Decomposition (MED) (*Eren et al., 2015a*), which uses Shannon entropy to identify highly variable nucleotide positions among amplicon sequences and iteratively decomposes a given sequencing dataset into oligotypes, or 'amplicon sequence variants' (ASVs), in which the entropy is minimal. The single-nucleotide resolution afforded by this strategy allows the identification of closely related but distinct taxa, better explaining micro-diversity patterns that may remain hidden otherwise (*Eren et al., 2014*; *Needham et al., 2017*). Reads were filtered to ensure only organisms grown during the culture period are presented in our data (see Materials and methods and *Figure 2—figure supplement 2*). Our data revealed no significant variation in the community composition between biological replicates or cultivation time for a given culture method (droplets or plates) and media (*Figure 2—figure supplement 3*). Across media and cultivation time, the community richness (number of detected ASVs) in droplets was larger than that on plates (*Figure 2a–b*,

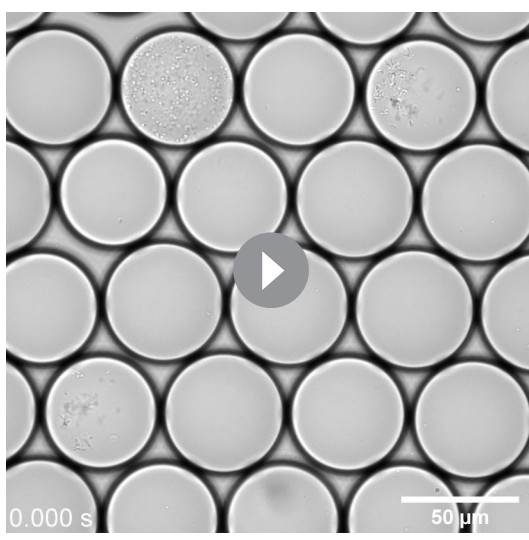

**Video 1.** Human stool bacteria cultured in BHIS droplets for 1 day.
https://elifesciences.org/articles/56998#video1

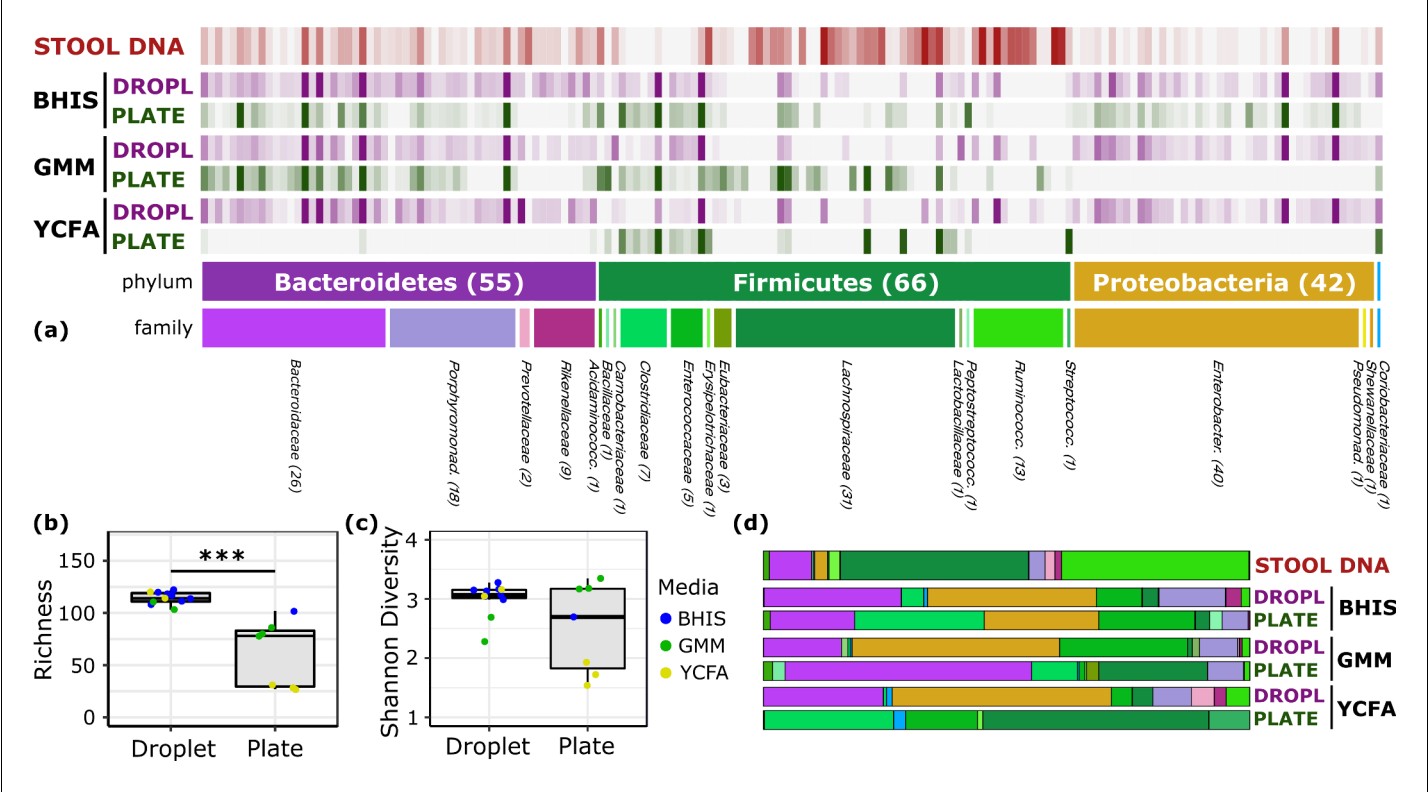

**Figure 2.** Comparison of human stool bacteria cultured on plates versus in droplets. (**a**) The relative abundance for each ASV organized by phylum and family is plotted for the raw stool and for representative droplet and plate cultures in each medium. The dark bars indicate a higher relative abundance. (**b**) The community richness averaged over cultivation time and media is increased in droplets over plates (p<0.005) but (**c**) the Shannon diversity is not. (**d**) Family-level relative abundance for representative droplet and plate cultures. The community composition between droplet cultures of different media is more similar than between plates.

The online version of this article includes the following figure supplement(s) for figure 2:

**Figure supplement 1.** A one day cultivation of the donor FMT stool sample in GMM medium.

**Figure supplement 2.** Amplicon sequence variant (ASV) filtering.

**Figure supplement 3.** Rank-abundance curves for independent experiments in BHIS, GMM, and YCFA medium cultivated in droplets or on plates.

**Figure supplement 4.** Intra-phyla (**a**) richness and (**b**) diversity for droplets (DR) and plates (PL) evaluated for Bacteroidetes (Bacter.), Firmicutes (Firmic.), and Proteobacteria (Proteo.).

**Figure supplement 5.** Relative abundance of the top five most abundant ASVs in each sample with taxonomic identification.

**Figure supplement 6.** Bray-Curtis hierarchical clustering of the family-level composition for all samples.

p<0.005, Mann-Whitney *U* test). In particular, droplets enabled an increase in richness between 15% (BHIS) and 410% (YCFA). The community diversity, measured by the Shannon index, increased in BHIS and YCFA droplets over their corresponding plates, but not in GMM (*Figure 2c*). The intra-phyla richness of across plates was non-normally distributed, with a notable lack of representation of Bacteroidetes on YCFA and Proteobacteria on GMM and YCFA (*Figure 2—figure supplement 4*). The most abundant ASV in all droplet samples, except 2 day GMM droplets, belonged within the closely related genera *Hafnia* and *Obesumbacterium* and had a mean abundance of 24% averaged across media and cultivation time (*Figure 2—figure supplement 5*). The droplets also featured a similar taxonomic composition at the family level across the three media, whereas the plate-based cultures more drastically differed from each other and from the input sample (*Figure 2d* and *Figure 2—figure supplement 6*).

One of the general bottlenecks of plate-based cultivation efforts is to detect and isolate organisms that are rare in the input sample, because abundant taxa are often over-represented in plates (*Zou et al., 2019*). Our data showed that droplets were able to grow a larger number of organisms that were low-abundance (<1%) in the original stool sample based on 16S rRNA gene amplicons

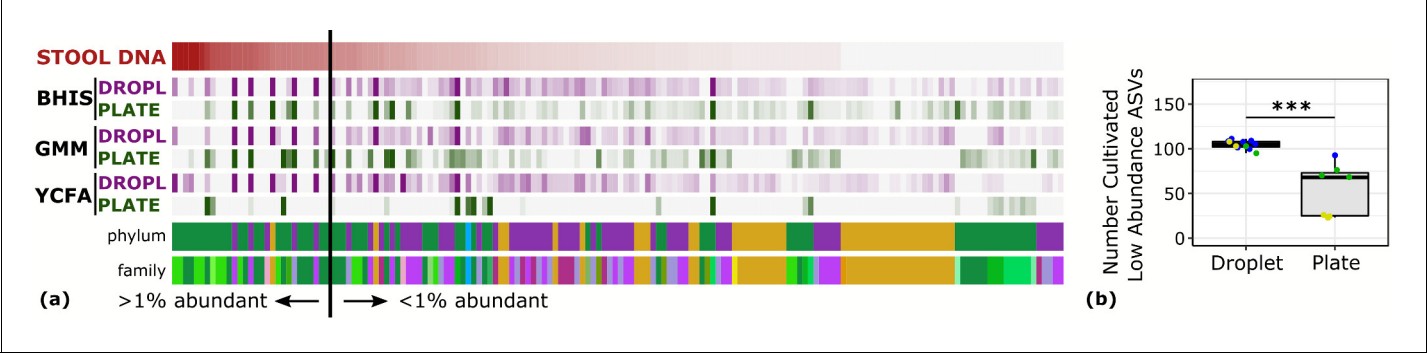

**Figure 3.** Droplet culture improves the cultivation of low abundant organisms. (**a**) The relative abundance per ASV organized by relative abundance in raw stool. The phylum and family colors correspond to the labels shown in a. (**b**) The number of cultivated low abundant ASVs in the raw stool sample (<1%, total of 130 ASVs) averaged over cultivation time and three different media is increased in droplets over plates. The legend for the phylum and family color labels is depicted in *Figure 2a*.

(*Figure 3*, p<0.005, Mann-Whitney *U* test). In particular, out of ASVs that were <1% abundant in the original stool sample, 105 ASVs from droplets and 57 ASVs from plates, averaged over cultivation time and media, were detected. Across droplet and plate experiments, 41 ASVs were detected which were not detected in the raw stool and would require an increased read count to resolve their true abundance. Next, we investigated how the composition of closely related taxa that resolve to the same taxonomic group in our cultivation efforts compared to their composition in the stool. For this, we performed an oligotyping analysis on all sequencing reads that matched to a single taxon, *Bacteroides*, an abundant genus in our dataset and one of the most variable genera across individuals (*Arumugam et al., 2011*). Hierarchical clustering of our samples based on the distribution of *Bacteroides* oligotypes revealed that the composition of *Bacteroides* populations measured by the 16S rRNA gene amplicons in droplet-based cultures were more similar to those in raw stool than plate-based cultures, as they clustered closer to the stool sample (*Figure 4*, p<0.005, multiscale bootstrap resampling). This indicates the influence of growth biases associated with plate-based cultivation was lessened in droplet-based cultures regardless of the medium, and a larger fraction of *Bacteroides* populations were accessible through droplets (*Figure 4*). In summary, droplet-based cultivation increases the richness and representation of rare taxa broadly across gut-associated phyla.

## Sorting slow growing organisms in droplets further amplify the abundance of rare taxa

Relatively slow growth rate is one possible explanation for the apparent low abundance of any given taxon within a sample. To investigate whether we could increase the relative abundance of ASVs which were <1% abundant in the raw stool sample, we automatically sorted droplets based on the colony density (*Video 2*). We performed two independent sorting experiments using human stool samples grown in BHIS droplets to keep only low-density colonies. The false positive sorting rate was low, with at most 8% of

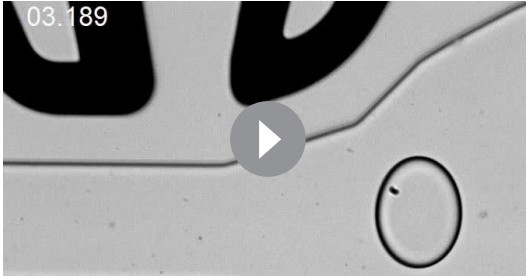

**Video 2.** Droplet sorting in an anaerobic environment for human stool bacteria cultured in BHIS droplets for 1 day. The top left time stamp is in seconds. For each droplet, the region between the 2 red dots is analyzed using the Wavelet OD. For empty droplets and droplets containing a dense colony, the Wavelet OD does not satisfy the thresholding criteria (the decision is labeled as false, 'F') and the droplets flow down the waste path. When the threshold criteria are met for a sparse colony (the decision is labeled as true, 'T'), the electrodes are actuated sending the droplet to the 'keep' path. We note that the spots in the oil phase were observed even for droplets generated without bacteria.
https://elifesciences.org/articles/56998#video2

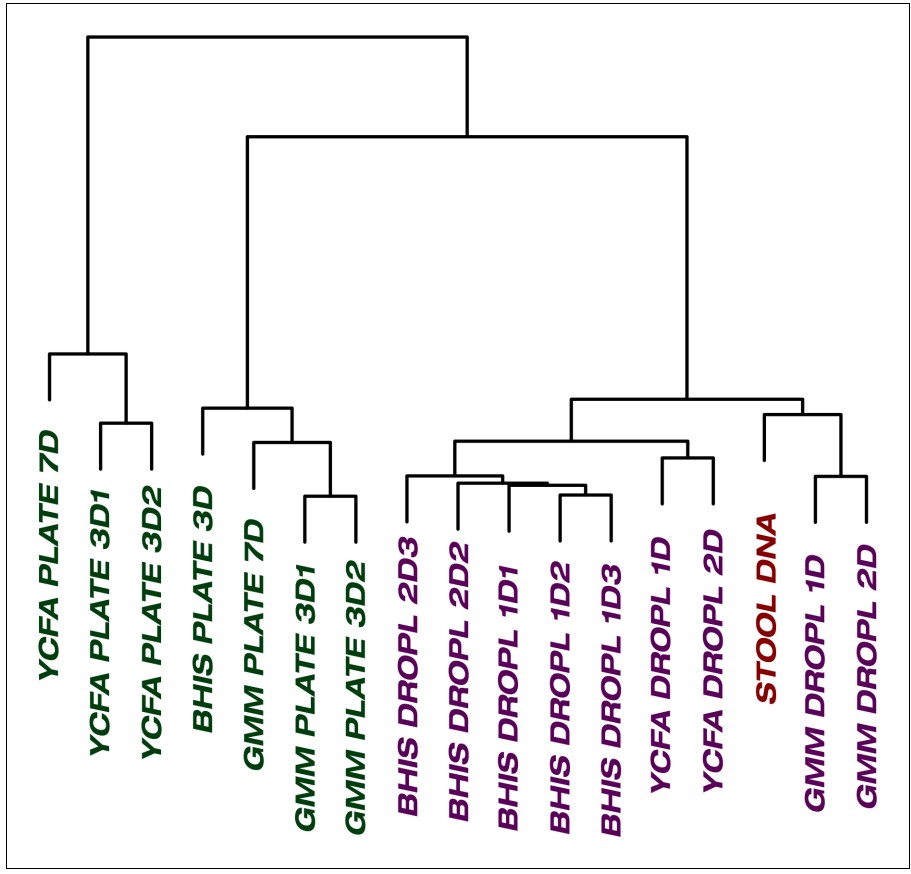

**Figure 4.** The culture of stool in droplets enables cultivation of clinically relevant *Bacteroides spp.* which did not grow on plate cultures. Hierarchical clustering of *Bacteroides* oligotypes across seven plate cultures and nine droplet cultures reveals a stronger association of droplet cultures towards the raw stool as compared to plates.

droplets incorrectly sent into the keep path. Sorted droplet cultures resulted in a shift in community composition (p<0.005, Kolmogorov-Smirnov test), with a noticeable change in the abundance of the top 20 ranked ASVs (*Figure 5a*). Next, we investigated which ASVs were amplified from <1% in raw stool to >1% in unsorted and sorted droplets. The average number of ASVs amplified from the <1% to >1% condition in unsorted BHIS droplets was 6.5 out of the 158 ASVs detected in total, while sorting increased the average number of amplified ASVs to 12.5 out of 158 (*Figure 5b*), thereby indicating that droplet sorting based on optical density can provide some preference in amplifying low abundance taxa. As a case in point, plate-based dilution cultures would require at least 65 standard culture plates to have a 90% chance at isolating the 6 *Alistipes* populations enhanced by the two droplet sorting experiments (see *Figure 5—figure supplement 1* for detailed explanation). Additionally, we note that sorted samples amplified taxa across a wider range of the phylogenetic tree than unsorted samples. Finally, to ensure that bacteria remain viable after droplet cultivation, we streaked sorted droplets from one experiment onto an agar plate and cultured the bacteria on the plate for 2 days. To test whether growing colonies on the plate represented distinct taxa, we randomly picked 24 of them. Sanger sequencing of their 16S rRNA genes resolved to genera *Hafnia* (12/24), *Enterococcus* (8/24), and *Bacteroides* (4/24) (*Figure 5—figure supplement 2*). In total, sorting slow-growing organisms in droplets leads to a further enhancement in the representation of rare taxa beyond that achieved by droplet isolation and cultivation alone.

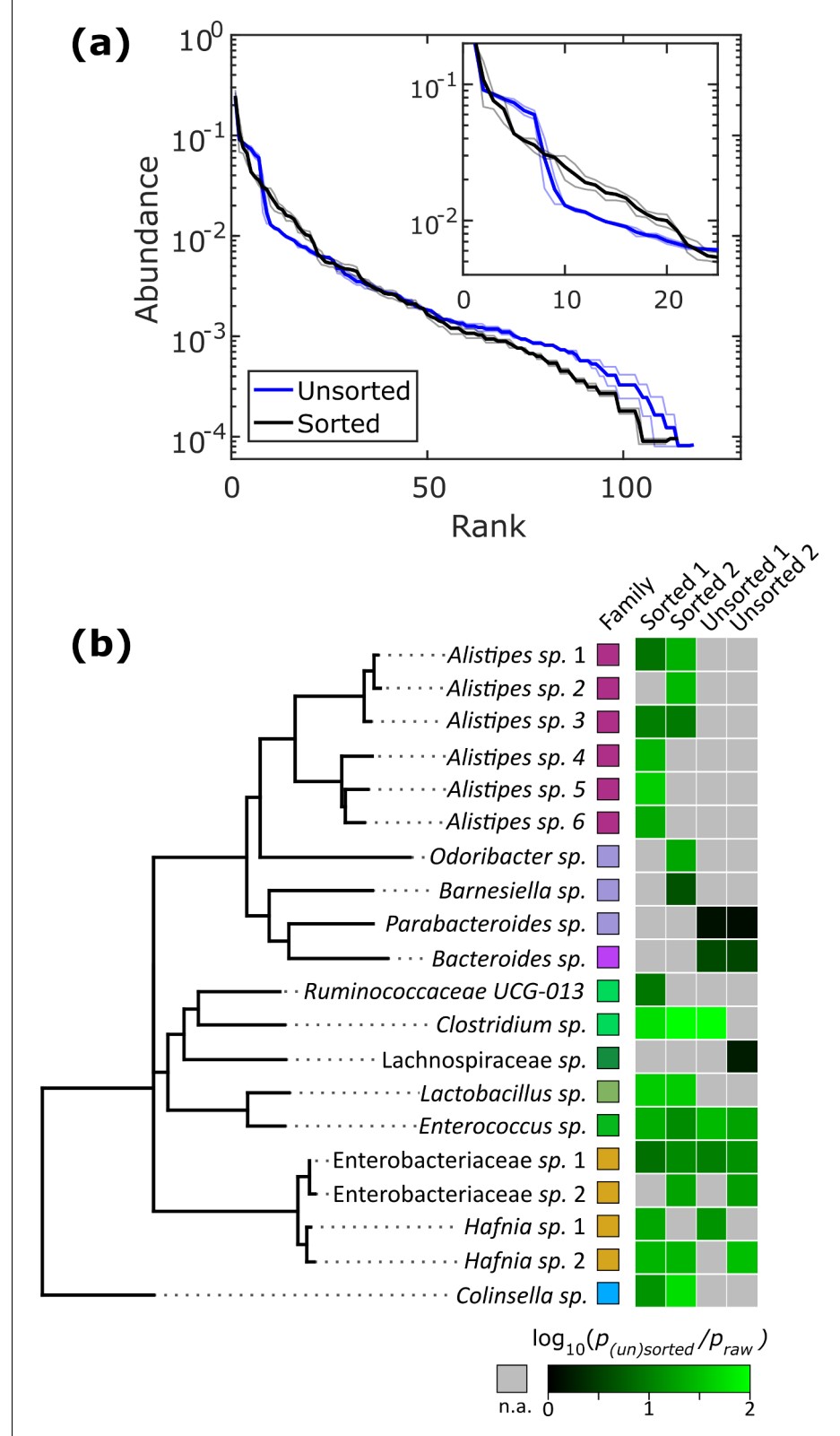

**Figure 5.** Isolation of slow-growing species from human stool microbiota. (**a**) The rank-abundance curves show that sorting based on colony density changes the overall community composition. A zoomed in portion of the rank-abundance curve is shown in the inset. (**b**) Phylogenetic tree of ASVs which were <1% abundant in raw stool but were increased to >1% in at least one sorting experiment. The ratio of amplification for strains amplified

*Figure 5 continued on next page*

*Figure 5 continued*

above the 1% limit is depicted in the heat map. N.A. (shown in gray) indicates ASVs which were not amplified from <1% to>1%. The legend for the family color labels is depicted in *Figure 2a*.

The online version of this article includes the following figure supplement(s) for figure 5:

**Figure supplement 1.** Comparison between droplets and plates for the isolation and cultivation of the 6 *Alistipes* populations enhanced by droplet sorting shown in *Figure 5b*.

**Figure supplement 2.** Bacteria can be cultured on traditional plates following cultivation in droplets.

## Antibiotic screening in droplets uncovers novel resistant members in fecal transplant microbiome

Rapid emergence and spread of antibiotic resistance is a major global public health problem, threatening prevention and treatment options for bacterial infections. Antibiotic resistance also poses a major risk for the development of new treatment methods for gastrointestinal tract diseases. For instance, *Clostridium difficile* is known to be resistant to multiple antibiotics, and FMT emerged as a major treatment option for recurrent *C. difficile* colitis (*Wang et al., 2016*). Therefore, antibiotic resistance screening of stool samples is essential for determination of healthy donors and minimizing health risks for large scale FMT applications.

In plate-based antibiotic screening, bacterial antibiotic resistance may be unobservable because the plate environment introduces an artificial bias that prevents that organism from growing. Since our droplet technology reduced biases associated with plates, and in particular increased the richness and representation of rare taxa, we hypothesized that our droplet technology could uncover antibiotic resistant members from the FMT donor stool which could not be determined using plate-based cultivation. We chose three widely used antibiotics – ampicillin (100 µg/mL), ciprofloxacin (5 µg/mL), and vancomycin (10 µg/mL), and screened for resistant members in plates and droplets (*Figure 6* and *Figure 6—figure supplement 1*). The concentrations were chosen to be commensurate with commercially available plates. In the case of droplet cultures, droplets without a growing colony were removed by sorting in order to remove dead or nonviable cells – the reads therefore represent only organisms which grew in the presence of the antibiotics.

Overall, droplets detected a much larger number of resistant organisms than plate-based cultures (*Figure 6—figure supplement 2*). In *Figure 6*, we show ASVs with the most significant difference between droplet and plate screens by filtering for ASVs which are at least 0.5% abundant and 10x greater than their counterpart. Overall, the droplet culture identified 21 new organisms from the human stool sample that exhibited strong antibiotic resistance. Additionally, ASVs which are present at greater than 1% abundance in both droplets and plates are depicted. Both droplets and plates recovered known resistance patterns including *Bacteroides spp.* and *Parabacteroides spp.* resistance to ampicillin (*Boente et al., 2010*), *Enterococcus sp.* and Enterobacteriaceae *gen. sp.* resistance to ciprofloxacin (*Hooper, 2002*; *Paterson, 2006*), and Enterobacteriaceae *gen. spp.* resistance to vancomycin (*Citron et al., 2012*). At the filtering criteria listed above, growth on plates exceeded droplets only for five populations across all three antibiotics, including a *Bacteroides sp.* on ampicillin plates and 4 *Clostridium spp.* on ciprofloxacin plates. Potentially, this preferred growth reflects an improved base fitness on antibiotic-free plates versus droplets for these organisms (*Figure 2a*). Droplet-based screening detected a number of important antibiotic resistant organisms (21 unique organisms) which could not be determined using plates. For example, ampicillin and vancomycin droplet-based screening detected growth of an organism within the genera *Pseudomonas*, which contains many opportunistic pathogens, and also within the genera *Shewenella*, which is a progenitor of antibiotic resistance genes in humans (*Yousfi et al., 2017*). Screening for ciprofloxacin resistance within droplets better displayed known resistances of *Bacteroides spp.* and *Parabacteroides spp.* to fluoroquinolone class antibiotics (*Snydman et al., 2017*) as well as the reduced activity against gram-positive bacteria (*Poole, 2000*). In summary, our droplet platform increases the detection of antibiotic resistant organisms present in human gut samples.

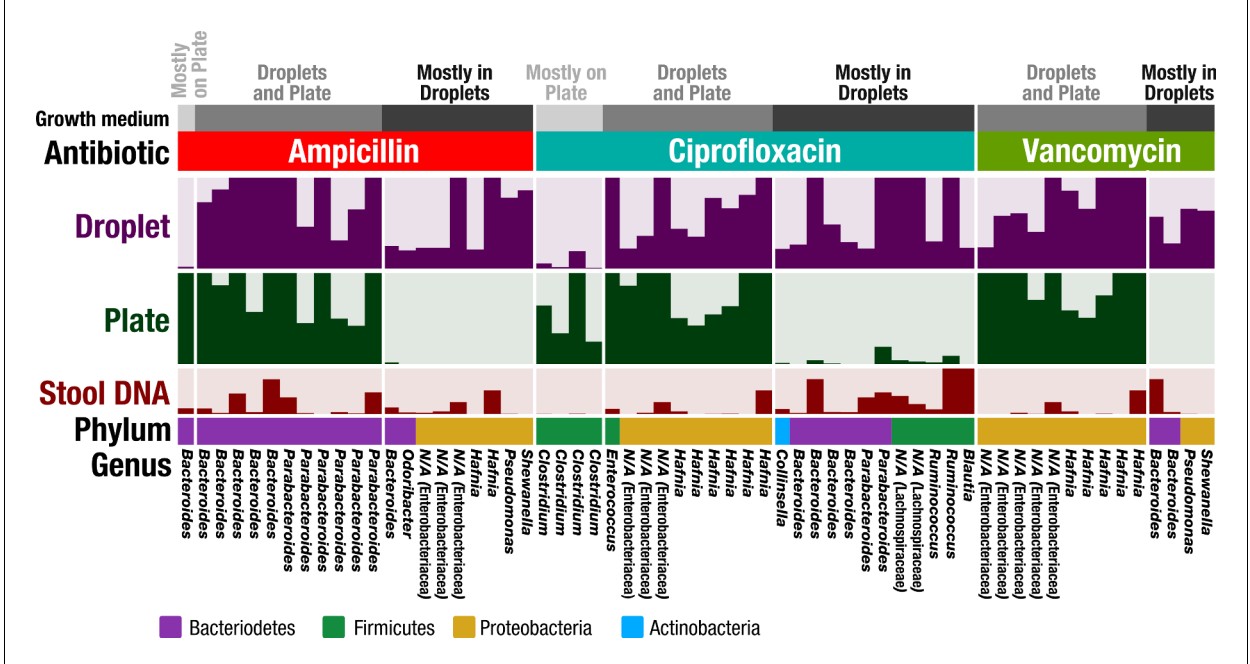

**Figure 6.** Antibiotic resistance screening of FMT donor stool in droplet cultures reveals antibiotic resistant members. The relative abundance of ASVs detected from droplet and plate cultures (rows) is shown for the three separate antibiotics along with the relative abundance in the raw stool sample. Each row is a linear scale between 0% and 2.5% relative abundance. For each antibiotic, ASVs are grouped according to plate preferred growth (light grey), droplet preferred growth (dark grey), or growth in droplets and plates (grey). Dead and nonviable cell DNA was removed via droplet sorting (for droplets only) and ASV filtering so that the ASVs depicted represent only organisms which grew in the presence of the antibiotics during the culture period. *N/A* indicates taxa which are unassigned at the genus level.

The online version of this article includes the following figure supplement(s) for figure 6:

**Figure supplement 1.** ASVs with taxonomic assignment in antibiotic droplet and plate experiments.

**Figure supplement 2.** Ecological measures of the cultivated community composition on droplets and plates in the presence of ampicillin (AMP, red), ciprofloxacin (CIPRO, blue), and vancomycin (VANCO, green) for the (**a**) richness, (**b**) the Shannon diversity, (**c**) number of ASVs which are <1% in the raw stool sample, and (**d**) rank-abundance.

## Discussion

Here, we developed a droplet-based microfluidic platform for isolating, cultivating, and sorting human gut-associated anaerobic bacteria. We cultivated the bacteria across three different rich media using our microfluidic platform and compared the growth to that on conventional plates. The droplet cultivation featured several advantages including an increase in community richness over plates by 15% up to 410%, depending on the medium, and an enhancement in the cultivation of low abundance (<1%) strains in raw stool. We also found a reduction in the variability of community composition across different media, and the droplets enabled cultivation of strains from the clinically important genus, *Bacteroides*, several of which did not grow on plates within the experiments conducted here. Further, sorting droplets based on colony density led to a further enhancement of low abundance strains in the sorted fraction. Finally, we applied our technology towards the detection of antibiotic resistant organisms present in FMT donor stool. Cultivation of the FMT sample in droplets led to the detection of many antibiotic resistant organisms which could not be detected using traditional plate cultures.

The droplet-based culture provides several key benefits over traditional plate-based cultivation. First, droplets offer a high-throughput platform for culturing, manipulating, and monitoring bacterial colonies. In aerobic systems, droplet microfluidics has enabled significant progress in fields ranging from pathogen detection, to antibiotic susceptibility testing, to strain engineering (*Kaminski et al., 2016*). Here, we demonstrated that droplet generation, cultivation, and sorting can be extended to anaerobic systems, potentially paving the way towards improved throughput in studying gut

microbiota. Second, parasitism, amensalism, and competition are eliminated between strains since each colony is isolated in its own droplet. Our data showed a broad representation of taxa across phyla in droplets (indicating that the base nutrient requirements are sufficient for growth), but a heterogeneous representation on plates. For instance, the lack of Bacteroidetes and Proteobacteria on YCFA plates suggests many of the Firmicutes members have a competitive advantage over the other phyla on YCFA. Although we are uncertain as to the exact cause, a comparison between experiments employing picking of thousands of colonies on YCFA medium versus 70 separate media finds a similar diversity of Firmicutes between the two studies even though only YCFA medium was used, suggesting a strong fitness of many Firmicutes on YCFA (*Forster et al., 2019*; *Lagier et al., 2016*). Competitive interactions on the YCFA plates may therefore have led to dominance of many Firmicutes, while the lack of competition in YCFA droplets enabled representation from the other phyla. This elimination of competition can also favor slow-growing organisms. For instance, our droplet sorting for slow-growing organisms enriched for six separate *Alistipes* populations (Rikenellaceae family, *Figure 5*). These *Alistipes* were either not detected or in very low abundance in plate cultures (*Figure 2*). Together, this suggests that these *Alistipes* are slow growing for all three tested media and that isolation in droplets prevents competitive overgrowth. Finally, the concentration of quorum sensing molecules can increase in droplets faster than bulk culture due to the inherently small volume of each droplet (*Boedicker et al., 2009*). Interestingly, the distribution and use of quorum sensing signaling genes in gut ecosystems may be wide spread – a recent study identified ~38% of genomes from the rumen microbiota possess quorum sensing-related genes (*Won et al., 2020*). Although we do not know the total extent to which quorum sensing is utilized by droplet-grown bacteria, one likely candidate is *Hafnia spp.* which were the predominant population in droplets across all three rich media and the growth dynamics of at least some strains of *Hafnia alvei* are modulated by quorum sensing (*Hou et al., 2017*). Combined, these droplet benefits – high-throughput, reduction of competition, and increase in quorum sensing – likely led to the improvement in droplet-grown representation. However, we note that disentangling the interplay of these benefits to determine the exact mechanism which allowed any given organism to grow would likely require mechanistic studies on a case-by-case basis.

As a case study, we applied our technology towards detecting antibiotic resistant members present in the FMT donor stool. Recently, transplantation of an FMT donor stool sample led to two recipient patients developing bacteremia, with one patient dying (*DeFilipp et al., 2019*). The stool sample was found to contain a rare extended-spectrum beta-lactamase producing (ESBL) *Escherichia coli* only after bacteremia occurred in the recipient patients. This rare ESBL *E. coli* went undetected during the initial safety screening of the donor stool. Here, we demonstrated a high-throughput method to detect rare antibiotic resistant organisms that might otherwise remain undetected by traditional cultivation. Our technology thus serves as an effective and efficient tool for screening FMT donor samples and their safer use in transplantation.

We envision that several further refinements to our droplet-based cultivation strategy could address limitations of the current study. First, in order for anaerobic droplet cultivation to be widely utilized among microbiologists, the droplet generating and sorting technologies must be easily integrated into standard microbiology workflows. Since microfluidic droplet generating devices are now commercially available (*Dolomite, 2020*), methods presented in our study are likely immediately transferable for isolation and cultivation of anaerobic bacteria in droplets. However, anaerobic droplet sorting is technologically more involved and will likely require commercial development of an anaerobic droplet sorter before it is widely adopted. Second, in this study, we isolated single living bacterial cells into droplets in order to prevent interspecies competition. However, isolation inhibits the growth of organisms that rely on other microbial or host cells, such as Saccharibacteria (formerly TM7) (*He et al., 2015*), or obligate endosymbionts, such as *Wolbachia* (*Hosokawa et al., 2010*). In droplets, co-encapsulation of two cross-feeding auxotrophic strains into a single droplet can induce growth, whereas growth will not occur when only one auxotroph is present within a droplet (*Park et al., 2011*; *Hsu et al., 2019*). Future studies could stochastically co-encapsulate multiple gut bacteria into droplets (by increasing the loading cell density during droplet generation) and investigate the resulting growth dynamics. Third, the arid environment of the anaerobic chamber led to a reduction in droplet volume over 4 days due to evaporation, limiting the extent of longitudinal studies. This issue could likely be resolved through humidity control inside the incubator. Finally, here we loaded droplets with three different rich media in order to broadly enrich the cultivated community

representation across taxa, since previous surveys found the majority of gut bacteria which grow in defined nutrient limited media also grow in rich media (*Tramontano et al., 2018*). However, some bacteria require minimal medium with specific carbohydrates, vitamins, or trace elements (*Tramontano et al., 2018*; *Oberhardt et al., 2015*) while others utilize surface features such as hydrophobicity, roughness, and surface chemistry to form biofilms and proliferate (*Tuson and Weibel, 2013*). Further enrichment of anaerobic organisms within our droplet platform could be achieved by incorporating droplet generation with defined medium (*Villa et al., 2020*), combinatorially generated gradients of medium (*Churski et al., 2012*), or varying the droplet surfactant chemistries to improve biofilm formation (*Chang et al., 2015*).

Our approach for isolation, cultivation, and sorting of gut microbiota in droplets enriched the representation of taxa across bacterial phyla, including organisms which are rare and/or slow growing. The improved representation of taxa afforded by droplet cultivation enabled the detection of antibiotic resistant organisms in an FMT donor stool sample which were not detected by traditional plate-based cultivation. Additionally, bacteria remained viable throughout the droplet cultivation and sorting processes suggesting that our anaerobic droplet technology is compatible with traditional downstream microbiology techniques. Going forward, our technology could facilitate overcoming difficulties in traditional plate-based cultivation and pave the way for rapid recovery and detection of novel strains in complex systems such as the human gut microbiome.

## Materials and methods

### Microfluidic device fabrication

The microfluidic droplet generation and droplet sorting devices were fabricated using soft lithography techniques. The device architectures were adapted from *Mazutis et al., 2013*. Briefly, we first fabricated molds from the negative photoresist, SU-8 3050, on 4" silicon wafers. The height of both the droplet generation and sorting devices was 50 μm. We then poured a 10:1 ratio of PDMS (RTV 615) parts A to B onto the mold, degassed the PDMS, and cured at 80℃ for at least 1 hr. Next, we removed the cured PDMS from the mold, punched holes for the inlets, outlets, and electrodes, and plasma bonded the PDMS to either a glass slide (for droplet generating devices) or a glass slide with a conductive indium-tin oxide on the rear side (for droplet sorting devices, Delta Technologies - Part No. CG-811N-S207). To increase the microchannel hydrophobicity, we coated the microchannels with Aquapel (Pittsburgh Glass Works) followed by Fluorinert FC-40 (Sigma). Finally, for droplet sorting devices, we created the electrodes by placing the microfluidic device on a 90℃ hotplate, flowing a low melting temperature solder (Indium Corporation of America, 51% In/32.5% Bi/16.5% Sn) into the electrode holes, and connecting the solder to standard wires.

### Microfluidic device control system

The syringe pumps, high-frame rate camera, and electrodes are controlled through custom written LabView code. We generated droplets using two syringe pumps (Harvard Apparatus Pump 11 Pico Plus Elite) which controlled the liquid and oil (Bio-Rad Droplet Generation Oil for EvaGreen) flow rates. We used two separate droplet generating devices here (see *Supplementary file 1*) with flow rates specified in the Source Data 1, Experiment Info. The droplet volumes ranged from ~65–115 pL. In a typical experiment, approximately 0.5–1 mL of droplets were generated in approximately 20 min – 1 hr, depending on the droplet generating device. For droplet sorting, the droplet reinjection flow rate was set to 20 μL/hr and the oil phase for droplet spacing was set to 180 μL/hr. The microfluidic devices were monitored using an inverted microscope (Nikon Ts2R) under 4x and 10x magnification. A high-frame rate camera (Basler acA640-750um) captured 672 × 360 pixel images at a rate of 925 Hz and the exposure time was set to 59 μs per frame. Our LabView code automatically analyzed each droplet near the sorting junction and made a sorting decision based off the wavelet OD (see droplet image analysis). Droplets which satisfied the sorting conditions were sent into the keep path by actuating the electrodes (*Figure 1d*). The remaining droplets flowed down the waste path. The sorting rate was ~30 Hz. The electrodes were actuated by outputting a true decision to an NI-DAQ 6211 which set the analog out to the desired voltage. The analog output voltage is then amplified (TREK Model 2220) at 200 V/V. The electrode actuations used a 10 kHz, 800 V p-p, 30% duty cycle square wave which was activated for 10 ms. An upper limit on the false positive sorting error,

$\varepsilon$, is given by $\varepsilon \leq [f^{un} (1 - f^s)] / [f^s (1 - f^{un})]$, where $f^{un}$ is the fraction of slow-growing colonies in the unsorted sample and $f^s$ is the fraction of slow-growing colonies in the sorted sample (i.e., the droplets sent into the keep path). The estimate on $\varepsilon$ is an equality when the false negative rate is zero. However, we did not count the slow-growing fraction in the waste stream and therefore the actual false positive rate is likely lower than the upper limit.

## Droplet image analysis

We sorted bacterial colonies in droplets based on an optical density-like measurement, which we termed the Wavelet OD. Custom LabView code first located each droplet as it approached the sorting junction by detecting the droplet edges along the center of the channel (i.e., the red dots in *Figure 1d*). The interior region of each droplet (~60×80 pixels) was then analyzed using a discrete wavelet frame decomposition from the LabView function IMAQ Extract Texture Feature VI (*Unser, 1995*). We optimized the wavelet parameters for speed and accuracy. In particular, we used biorthogonal 3.1 wavelets with the Low Low High subband, a 15 × 15 pixel window with a step size of 5 pixels, and the co-occurrence matrix quantized into 15 gray levels with a 3 × 3 pixel displacement distance. The number of non-zero elements in the wavelet feature vector was then normalized to one to obtain the Wavelet OD. Droplets with a Wavelet OD between 0.3 and 0.7 were empirically identified as slow-growers and sorted into the keep channel. Droplets with a wavelet OD of less than 0.3 were typically empty droplets while droplets with a wavelet OD of greater than 0.7 contained a dense bacteria colony.

## Anaerobic chamber

Bacteria cultivation, droplet generation, and droplet sorting were all carried out inside a vinyl anaerobic chamber (Coy Laboratory Products) supplied with an 86% $N_2$/10% $O_2$/4% $H_2$ gas mixture. The $O_2$ and $H_2$ concentrations were monitored using an anaerobic monitor (Coy CAM-12). The $H_2$ concentration was maintained between 1.5–2.5% and the $O_2$ concentration was typically less than one ppm. A hydrogen sulfide reducing column (Coy) was placed inside the chamber to prevent corrosion of the electronic components from $H_2S$ buildup.

## Media

### BHIS

We first mixed 500 mL water, 18.5 g Brain Heart Infusion Broth (Sigma), 2.5 g yeast extract, 0.25 g L-cysteine, and 7.5 g agar, if making plates. We then autoclaved the solution and added 0.5 mL of 0.1% Vitamin K solution in 95% ethanol and 0.5 mL of a filter-sterilized hemin solution of 5 mg/mL in 0.1 M NaOH.

### BHIS-ASF

For ASF cultures, we modified the BHIS medium described above by adding 5% newborn calf serum, 5% sheep serum, and 5% horse serum (all sera from Fisher Scientific).

### GMM

GMM was prepared following the directions outlined by *Goodman et al., 2011*.

### YCFA

We used the following modified version of DSMZ media 1611. We first mixed 10 g casitone, 2.5 g yeast extract, 5 g dextrose (D-glucose), 0.045 g $MgSO_4 \times 7 H_2O$, 0.09 g $CaCl_2 \times 2 H_2O$, 0.45 g $K_2HPO_4$, 0.45 g $KH_2PO_4$, 0.9 g NaCl, 0.001 g resazurin sodium salt, 1.9 mL acetic acid, 0.7 mL propionic acid, 90 μL iso-butyric acid, 100 μL n-valeric acid, 100 μL iso-valeric acid, and 500 mL DI water. The medium was then boiled for 10 min while stirring and then cooled. Next, we added 4 g $NaHCO_3$, 0.5 g L-cysteine-HCl, 0.01 g hemin, 0.0025 g mucin, and 15 g agar, if making plates. DI water was added so that the total volume becomes 990 mL, the pH was adjusted to 6.7–6.8, and the solution was autoclaved. After autoclaving, we added 10 mL of filter-sterilized vitamin solution. One liter of vitamin solution is made by mixing 2 mg biotin, 2 mg folic acid, 12 mg pyridoxine-HCl, 4.5 mg thiamine-HCl, 5 mg riboflavin, 5 mg nicotinic acid, 5 mg D-Ca-pantothenate, 0.1 mg vitamin B12, 5 mg p-aminobenzoic acid, 5 mg DL-thioctic acid, and 1L DI water.

## Human stool sample collection and droplet cultivation

The stool sample we used was previously collected from a fecal microbiota transplant donor (*Lee et al., 2017*). University of Chicago Ethics Committee and the University of Chicago Institutional Review Board (IRB 132–0212) approved the sample collection, and we obtained written and informed consent from the single stool donor. We aliquoted the sample by spinning down 50 µL of stool diluted in 100 µL of PBS, carrying forward the supernatant. The supernatant was stored at −80˚ C. The live cell and dead cell densities in our aliquots were measured to be ~1.5 ± 1.0 x $10^8$/mL and ~21 ± 3 x $10^8$/mL, respectively, using the Live/Dead BacLight kit (ThermoFisher). Because the fluorescent dyes in the Live/Dead BacLight kit require oxygenation of the surrounding medium to fluoresce, we exposed the aliquot to air, which may impact the true live/dead measurement. For plate cultures, 100 µL of the aliquoted bacteria suspension was spread onto the plates followed by scraping the plate after cultivation to collect DNA. For both plate and droplet cultures, the DNA was extracted using the Qiagen DNeasy Blood and Tissue Kit according to the manufacturer's directions. We chose a high plating density of ~1 cell/500 µm$^2$ – which resulted in 'bacterial lawns' after cultivation on rich medium. A high plating density was chosen to ensure rare cells are plated, as compared to limiting dilution where distinct colonies can be grown at the expense of not plating rare populations. For droplets, aliquots were diluted 200x, so that according to Poisson statistics, the mean percentage of droplets that will contain one living bacteria is 2% to 12%, accounting for the uncertainty in cell density and the different droplet volumes. In a typical experiment, we generated and cultured a ~0.5–1 mL emulsion of ~65–115 pL droplets (i.e.,~4–15 million droplets). The droplets were stored and cultivated in a snap-cap 15 mL culture tube (Fisherbrand - 149569C). After cultivation, we pipetted out 100 µL of the droplet emulsion (with an exception for sorted droplets described below), mixed it with 100 µL of 1H,1H,2H,2H-Perfluoro-1-octanol (PFO), vortexed and centrifuged the solution, and removed the oil and PFO. Because our sorting rate was limited to ~30 Hz, the total volume of the droplets which were sorted into the 'keep' path was significantly less than 100 µL. In the two sorting experiments conducted here, the sorted droplets only formed a very thin layer on top of the oil. Roughly, the estimated total volume observed by pipetting, was 1–10 µL. To extract the DNA, we therefore first added 150 µL DNA-free water and 150 µL PFO to the sorted droplets followed by the same vortex and centrifugation steps. For both plate and droplet cultures, we note that dead cells present in the initial inoculum may generate a small uncertainty in the detected cultivated organisms.

In order to verify that the droplet cultivation platform is also compatible with traditional microbiology workflows, we further cultured bacteria grown in droplets on a plate after sorting. In one experiment, we cultured human stool bacteria in BHIS droplets for 1 day, sorted the droplets, and then streaked ~10 µL of the sorted droplet emulsion and oil onto a BHIS plate. We then cultured the plate for 2 days at 37˚C and then randomly picked 24 colonies. Each colony was placed into 2 mL of BHIS broth, mixed, and 1 mL was extracted for Sanger sequencing. DNA was isolated using the Qiagen DNeasy Blood and Tissue Kit according to the manufacturer's instructions. We amplified the 16S rRNA using the primers 27F (AGAGTTTGATCMTGGCTCAG) and 1492R (GGTTACCTTGTTACGACTT) by mixing 250 nM of the 27F and 1492R primers, 10 µL GoTaq Green Master Mix (Promega), 2 µL extracted DNA, and 7 µL DNA free water followed by running PCR amplification with (i) 3 min at 95˚C, (ii) 30 cycles of 30 s at 95˚C, 30 s at 54˚C, 1 min at 72˚C, and (iii) 10 min at 72˚C. Each sample was then run through a 2% agarose gel, the band was excised from the gel, and the DNA purified using the QIAquick Gel Extraction Kit (Qiagen) according to the manufacturer's instructions. The DNA was then Sanger sequenced with the 27F and 1492R primers at the University of Chicago Comprehensive Cancer Center DNA Sequencing and Genotyping Facility. Finally, the forward and reverse reads were aligned in Benchling (https://benchling.com) using the MAFFT algorithm with default parameters and the consensus sequence was searched using the Standard Nucleotide BLAST for the closest taxonomic match.

## Antibiotic screening

Antibiotic resistance screening was performed by adding ampicillin (100 µg/mL), ciprofloxacin (5 µg/mL), or vancomycin (10 µg/mL) to BHIS plates or BHIS droplets. For antibiotic plate cultures, the raw stool inoculum was diluted 10,000x before plating (plating density of ~0.2 cell/mm$^2$). The droplet loading density was the same as in the rich media experiments. Plate screenings were cultured for 3

days and the DNA was then collected through a plate scraping. Droplet screenings were first cultured for 1 day. Next, because the addition of the antibiotics leads to more droplets containing nonviable single cells, inclusion of these nonviable cells would decrease our signal-to-noise. To prevent this, we therefore sorted droplets after the 1 day culture period using our optical density-based droplet sorter to remove drops without a grown colony. After sorting, the DNA was prepared for sequencing as described above.

## Sequencing

The Environmental Sample Preparation and Sequencing Facility at Argonne National Laboratory (Argonne, IL, USA) performed library preparation and sequencing of our DNA isolates following their established protocol developed through the Earth Microbiome Project (https://earthmicrobiome. org/protocols-and-standards/16s/). Briefly, 35 cycles of amplification were performed using the primer set described previously (*Caporaso et al., 2012*; *Caporaso et al., 2011*) that target the V4 region of the 16S rRNA gene to generate our amplicons from purified DNA, and Illumina MiSeq paired-end sequencing (2 × 151) was used to sequence our amplicon libraries. Although here we sequenced the V4 region, we note that sequencing of the V4-V5 region can improve taxonomic resolution (*Nelson et al., 2014*). We analyzed the raw sequencing reads using illumine-utils (*Eren et al., 2013a*) to (1) de-multiplexed raw sequencing reads into samples, (2) join paired-end sequences, and (3) remove low-quality sequences by requiring a minimum overlap size of 45 nucleotides between the two reads in each pair and removing any read that contained more than two mismatches in the overlapped region (mismatches in sequences survive these criteria were resolved with the use of the higher quality base). Finally, we inferred amplicon sequence variants (ASVs) in our dataset using Minimum Entropy Decomposition (MED) (*Eren et al., 2015a*) through the oligotyping pipeline v2.1 (*Eren et al., 2013b*), and taxonomy was assigned to each ASV using the SILVA database (*Quast et al., 2013*).

## Amplicon sequence variant filtering

For both droplet and plate cultures, DNA from dead or nonviable cells in the initial inoculum can be carried over into the collected DNA post culture. Therefore, to ensure that the measured ASVs represent organisms which grew during the culture period, we applied a conservative filtering threshold (*Figure 2—figure supplement 2*). We first created a threshold for each sample by fitting a percentage of dead or nonviable DNA, $p_{n.v.}$, from the raw stool initial inoculum to sample ASVs which were at least 10 times less than the corresponding raw stool ASVs. Thus, $p_{n.v}$ represents the fraction of dead or nonviable DNA in the post culture DNA. Next, a 90% confidence interval for each ASV was determined using maximum likelihood estimation (MLE) as follows. Consider a given ASV, $k$, with a true proportional abundance, $p_k$, and measured proportional abundance, $\hat{p}_k = x/n$, where $x$ is the number of ASV reads and $n$ is the total number of reads in the sample. Let $p_k$ be reparametrized by $\theta_k = \ln p_k / (1 - p_k)$. The 90% confidence interval for $\theta_k$ is given by $\hat{\theta}_k \pm 1.645 / \sqrt{-l''\left(\hat{\theta}_k; x\right)}$, where $l(\theta_k;\ x) = \mathrm{constant} + x\theta_k - n\ln\left(1 + e^{\theta_k}\right)$, is the reparametrized log likelihood estimator for the binomial process of random read sampling, and the derivatives are taken with respect to $\theta_k$, and evaluated at $\hat{\theta}_k$. The 90% confidence interval on the number of counts for ASV, $k$, is then given by transforming back to the lower and upper estimate of $p_k$ and multiplying by the number of sample counts. For each ASV, if the lower limit of the 90% confidence interval is below the fitted threshold, the read is discarded.

## Data analysis

We characterized each sample using standard ecological metrics including richness ($R$), Shannon's diversity index ($H'$), and rank-abundance curves. ASVs were first normalized by proportion so that for each sample, $\Sigma_i p_i = 1$, where $p_i$ is the proportional abundance of ASV $i$. The richness is the total count of ASVs detected within a sample and Shannon's index, $H' = -\Sigma_i\ p_i \log(p_i)$. Rank-abundance curves were obtained by ordering the ASVs by decreasing $p_i$ for each sample. Additionally, we also counted the richness of ASVs for each sample which were <1% abundant in the raw stool sample ($R_{low}$) in order to investigate if droplets can enhance the cultivation of rare species. We statistically tested the metrics $R$, $H'$, and $R_{low}$ using the nonparametric Mann-Whitney $U$ test under the null

hypothesis that the difference between the means of the metrics on plates and droplets, irrespective of culture media, antibiotic, and cultivation time, is zero. The statistical testing was implemented in the R package using the function wilcox.test. We also statistically tested if there was any difference in the rank-abundance distributions between samples with the same cultivation condition (i.e., droplet or plate) and the same medium. In particular, we applied the Kolmogorov-Smirnov test (R function ks.test) under the null hypothesis that two samples can be generated from the same distribution. Next, hierarchical clustering was applied to infer associations between samples. Samples were clustered at the family level by calculating the Bray-Curtis dissimilarity index in the R function vegdist in the package vegan and then plotting the dendrogram. Finally, we statistically tested that *Bacteroides* oligotypes cluster closer to raw stool using hierarchical cluster analysis with multiscale bootstrap resampling (R function pvclust) (*Suzuki and Shimodaira, 2006*).

We also tested if sorting slow-growing colonies could amplify the relative abundance of rare ASVs. Rank-abundance curves for the two independent sorting experiments were first generated and the combined sorted and unsorted distributions were statistically compared using the two-sample Kolmogorov-Smirnov test, which tests if two sample distributions can be drawn from a common distribution. Next, we investigated the ability of the sorted droplets to increase the abundance of low-abundant ASVs from the raw stool. We arbitrarily set the limit to 1%; ASVs which were <1% abundant in raw stool were considered amplified if the ASV's relative abundance was >1% in the sorted or unsorted droplets. The ASVs which satisfied this condition across two separate sorting experiments in BHIS droplets were pooled together for phylogenetic analysis. The phylogenetic tree was generated by performing multiple sequence alignment using the default settings on Clustal Omega followed by calculating a DNA Neighbour Joining tree in Jalview. Finally, the increase in proportional abundance for the ASVs which satisfied the above condition in unsorted and sorted droplets was calculated.

We used anvi'o v5.5 (*Eren et al., 2015b*) to visualize heat map visualizations of ASV percent relative abundances and clustering dendrograms, and used the open-source vector graphics editor Inkscape (available from https://inkscape.org) to finalize them for publication.

## Acknowledgements

We thank Dr. Michael Wannemuehler for generously providing the eight pure ASF strains and for notes on culturing the ASF, the Environmental Sample Preparation and Sequencing Facility at Argonne National Laboratory, the Pritzker Nanofabrication Facility, and the University of Chicago Comprehensive Cancer Center DNA Sequencing and Genotyping Facility. This research is supported by the NIDDK P30 University of Chicago Digestive Disease Research Core Center (DK42086), T32 DK07074 (supporting WJW), RC2 DK122394-01, The Samuel and Emma Winters Foundation Grant (2018–2019), and the GI Research Foundation of Chicago (supporting WJW). We thank the Mutchnik Family Fund (who supported AME and ARW) and the Duchossois Family Institute at the University of Chicago (for pilot funding to WJW, MT, AME, and ST).

## Additional information

### Competing interests
Savaş Tay: Savaş Tay is a founder and equity holder of BiomeSense Inc. The other authors declare that no competing interests exist.

### Funding

| Funder | Grant reference number | Author |
| --- | --- | --- |
| National Institute of Diabetes and Digestive and Kidney Diseases | DK42086 | Eugene B Chang |
| National Institute of Diabetes and Digestive and Kidney Diseases | RC2 DK122394-01 | Eugene B Chang |

| | | |
|---|---|---|
| Samuel and Emma Winters Foundation | 2018-2019 | Melikhan Tanyeri |
| GI Research Foundation of Chicago | | William J Watterson |
| The Mutchnik Family Fund | | A Murat Eren<br>Andrea R Watson |
| National Institute of Diabetes and Digestive and Kidney Diseases | T32 DK07074 | William J Watterson |
| Duchossois Family Institute at the University of Chicago | | Savaş Tay |

The funders had no role in study design, data collection and interpretation, or the decision to submit the work for publication.

## Author contributions

William J Watterson, Conceptualization, Software, Formal analysis, Funding acquisition, Validation, Investigation, Visualization, Methodology, Writing - original draft, Writing - review and editing; Melikhan Tanyeri, Conceptualization, Resources, Funding acquisition, Investigation, Methodology, Writing - review and editing; Andrea R Watson, Resources, Investigation, Methodology, Writing - review and editing; Candace M Cham, Yue Shan, Resources, Methodology, Writing - review and editing; Eugene B Chang, Conceptualization, Supervision, Funding acquisition, Project administration, Writing - review and editing; A Murat Eren, Conceptualization, Software, Formal analysis, Supervision, Funding acquisition, Validation, Investigation, Visualization, Methodology, Project administration, Writing - review and editing; Savaş Tay, Conceptualization, Resources, Supervision, Funding acquisition, Validation, Visualization, Methodology, Project administration, Writing - review and editing

## Author ORCIDs

William J Watterson https://orcid.org/0000-0002-5065-9634
Andrea R Watson http://orcid.org/0000-0003-0128-6795
A Murat Eren https://orcid.org/0000-0001-9013-4827
Savaş Tay https://orcid.org/0000-0002-1912-6020

## Decision letter and Author response

Decision letter https://doi.org/10.7554/eLife.56998.sa1
Author response https://doi.org/10.7554/eLife.56998.sa2

# Additional files

## Supplementary files

• Source code 1. ASV filtering to remove nonviable or dead cell DNA from reads.

• Source data 1. Experiment_Info. The droplet generation and cultivation information is listed for each experiment. 2. ASV_Counts_Matrix. The counts matrix for each ASV determined by Minimum Entropy Decomposition (MED) for the raw human stool sample and the human stool cultured on plates and in droplets. 3. ASV_Filtered_Percent_Matrix. Percent matrix for each ASV obtained by filtering the reads using the Matlab analysis file ASV_Filtering.m. 4. Sequences_Taxonomy. Closest taxonomic assignment of each ASV from raw and cultivated human stool samples. 5. Sequences. Sequences from MED analysis for raw and cultivated human stool samples. 6. Ecological_Measures. Richness, Diversity, Dominance, and Number Rare measurements for each sample and the intra-phyla richness and diversity across all 4 detected phyla. 7. Oligotyping_Percent_Matrix. Percent matrix for Bacteroides oligotype assignment for each sample. 8. Oligotyping_Representatives. Oligotypes of Bacteroides spp. 9. Sorting_Enhancement_Ratio. Enhancement ratio for ASVs which were <1% abundant in the raw stool sample but >1% in the droplet sorting experiments. 10. Sanger_Consensus_Sequences. The consensus sequence for each of 24 randomly picked colonies from sorted

droplets which were streaked onto a plate. For each sequence, the closest BLAST taxonomic assignment along with percent identity is also listed.

- Supplementary file 1. CAD file for microfluidic droplet generating and sorting devices.
- Transparent reporting form

### Data availability

All data generated or analysed during this study are included in the manuscript and supporting files.

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
