## [Decision Letter]

**Acceptance summary:**

This paper describes a new method that uses microfluidics to isolate and cultivate microbial cells in picoliter droplets and then sorts them by colony density. This method avoids many biases introduced by bulk culture and broadens sampling of diversity, as seen when applied to samples from the human gut microbiome. Using this method can reveal more taxonomic and functional diversity, such as antibiotic resistant organisms that are missed by bulk screening.

**Decision letter after peer review:**

Thank you for submitting your article "Droplet-based high-throughput cultivation for accurate screening of antibiotic resistant gut microbes" for consideration by *eLife*. Your article has been reviewed by three peer reviewers, including Vaughn S Cooper as the Reviewing Editor and Reviewer #1, and the evaluation has been overseen by Gisela Storz as the Senior Editor. The following individuals involved in review of your submission have agreed to reveal their identity: Babak Momeni (Reviewer #2); Hyun Jung Kim (Reviewer #3).

The reviewers have discussed the reviews with one another and the Reviewing Editor has drafted this decision to help you prepare a revised submission.

Summary:

All reviewers agree that this is an innovative method with promise for significant conceptual advances. The presentation is strong and the application to the prevalence of antimicrobial resistance is appropriate and demonstrates effects of population subdivision on resistance metrics.

Essential revisions:

Two issues merit further consideration: 1) One promise of the method is to explore the disconnect between plate counts and cell counts, but the manuscript does not address this problem as well as the data might permit. Reviewer 1 states this opportunity clearly. 2) The growth media used were nutrient-rich and likely less suited for culturing organisms adapted to low-nutrient conditions, as reviewers point out, and this potential shortcoming must to be considered.

Reviewer #1:

The problem of most microbiology is that the scale at which we study microbes is vastly larger than the scale at which they interact with one another. The use of droplets to partition and isolate mixed populations offers a powerful approach to study more realistic interactions at a relevant scale. This study is well grounded in microbial ecology, well written, and applies an innovative method and appropriate analyses of the system to an important question – how prevalent are antibiotic-resistant organisms within gut communities?

I also appreciate the use of multiple media types for cultivation and drugs as challenges for resistance. I don't have any significant criticisms.

Reviewer #2:

The authors propose massive parallel cultivation of strains from anaerobic gut microbiota using a droplet fluidic platform as an alternative to traditional cultivation of cells on plates. The main advantages of this approach is that individual cells are isolated, thus allowing the rare and/or slow-growing strains to grow even when abundant and fast-growing types are present in the original sample. In my opinion, the paper is clear, accessible, and of broad interest for microbiota studies. The proposed methodology also has the potential to be broadly adapted by other labs.

1) My only major comment is that the paper for the most part stays at the level of descriptive presentation of their observations, even though there is an opportunity to add explanations and generate insights based on what they have found. My main suggestion for this paper is the inclusion of a systematic discussion on potential origins of the differences between plating and droplets for cultivating different strains. The authors briefly allude to this in the second paragraph of their Discussion, but in my opinion, this is one of the primary contributions of this manuscript and deserves more in-depth analysis. In a simplified view, four possible mechanisms can be responsible for a strain's failure to grow: (1) absence of a required nutrient in the medium; (2) absence of an obligatory partner; (3) lack of accumulation of signal for quorum sensing (or in general, density dependent growth); and (4) inhibition of rare/slow growing strains by common/fast-growing strains either through exploitative competition (i.e. depletion of resources) or interference competition (e.g. production of toxins or inhibitory waste). The first one would be shared between plates and droplets (can be seen as an intrinsic limit of cultivation methods); the second one would be exacerbated in droplets; and the third and fourth ones could be potentially resolved in droplets. I think the authors can go into some more detail and make speculations about which one is more likely (based on their data), what are the main experimental parameters (e.g. culture volume, number of initial cells per culture,…) affecting each, and potentially the intrinsic or practical limits associated with each. This does not require any additional experiments necessarily, and the rich dataset that the authors have already produced should be sufficient to draw the necessary conclusions.

Reviewer #3:

The authors devised a microfluidics-based system that can isolate a single bacterial cell anaerobically. The microfluidic device in the system generates droplets that can encapsulate a single bacterium, and by integration with an imaging platform, the authors were able to sort out false-negative or -positive droplets. The authors used a fecal microbiome in the system to separate single bacterial cells and cultured the cells in the droplets. Because this technology enabled to encapsulate slow-growing bacteria in a single-cell level, the bacterial population could be more diverse after the culture in each droplet compared to the conventional plate culture. This confinement effect should be considerably applied when microbiologists attempt to grow the multi-species microbial consortia in vitro, thus the authors decently demonstrated the implementability of this confinement approach. Ultimately, the authors identified 21 novel antibiotic-resistant bacteria from the human stool sample, which could be further utilized in clinical practices. The work is technically sound and well-designed. However, a couple of concerns raised in this manuscript should be resolved by the author's clarification for the publication in *eLife*.

– It is unclear if the individual picoliter droplet really contains a single bacterium. The authors claimed the single-bacterium confinement in a droplet, but they never showed an initial entrapment of the bacterium as a single cell level, whereas authors only showed the droplets that contain fully grown bacterial cells. A simple bacterial Live/dead staining can be applied to clearly show the single-cell deposition per droplet.

– This study showed a unique anaerobic approach and promising applicability to improve the scalable, droplet-based microfluidics toward the culturomics. However, this approach still heavily relies on the nutrient-rich culture medium that may compromise the rare bacterial cells that require a minimal medium, particular trace elements, or host-derived metabolites. The main reason that the authors could find the higher richness and diversity in the droplet-based isolation compared to the agar plate method is simply that the system more accurately manages the cultured cells in each droplet under the confinement effect and the segregation of bacterial cells to each other, but not because they used a fancy medium. Thus, this method is still limited to claim to culture the unculturable fecal bacteria in vitro. This part should be substantially discussed.

---

## [Author Response]

Essential revisions:Two issues merit further consideration: 1) One promise of the method is to explore the disconnect between plate counts and cell counts, but the manuscript does not address this problem as well as the data might permit. Reviewer 1 states this opportunity clearly. 2) The growth media used were nutrient-rich and likely less suited for culturing organisms adapted to low-nutrient conditions, as reviewers point out, and this potential shortcoming must to be considered.

We appreciate the importance of both points and our revision includes new sections and edits to discuss them in greater detail. Point #2 does indeed hit upon the salient point that not all organisms are cultivable in rich media. We have now updated the Discussion and added citations to reflect this point (explained in greater detail in response to reviewer #3 Major Comment). In contrast to Point #2, fully addressing point #1 is difficult and will require in-depth, even perhaps organism-by-organism investigations due to multiple potential mechanistic explanations for the difference between plate cell counts and droplets. Our responses below (in particular the response to reviewer #2) and changes in the revised manuscript now elucidate these limitations further. We hope the editors and our reviewers will agree with our resolution.

Reviewer #2:[…] 1) My only major comment is that the paper for the most part stays at the level of descriptive presentation of their observations, even though there is an opportunity to add explanations and generate insights based on what they have found. My main suggestion for this paper is the inclusion of a systematic discussion on potential origins of the differences between plating and droplets for cultivating different strains. The authors briefly allude to this in the second paragraph of their Discussion, but in my opinion, this is one of the primary contributions of this manuscript and deserves more in-depth analysis. In a simplified view, four possible mechanisms can be responsible for a strain's failure to grow: (1) absence of a required nutrient in the medium; (2) absence of an obligatory partner; (3) lack of accumulation of signal for quorum sensing (or in general, density dependent growth); and (4) inhibition of rare/slow growing strains by common/fast-growing strains either through exploitative competition (i.e. depletion of resources) or interference competition (e.g. production of toxins or inhibitory waste). The first one would be shared between plates and droplets (can be seen as an intrinsic limit of cultivation methods); the second one would be exacerbated in droplets; and the third and fourth ones could be potentially resolved in droplets. I think the authors can go into some more detail and make speculations about which one is more likely (based on their data), what are the main experimental parameters (e.g. culture volume, number of initial cells per culture,…) affecting each, and potentially the intrinsic or practical limits associated with each. This does not require any additional experiments necessarily, and the rich dataset that the authors have already produced should be sufficient to draw the necessary conclusions.

The reviewer raises a very interesting point and one of the main questions we had in analysing our data – *why* do the droplets perform better in culturing organisms? As the reviewer correctly points out, there are four competing mechanisms which influence whether a given organism can grow or not. While we fully agree that these questions deserve much more attention, for reasons we explain below, teasing these factors apart *definitively and mechanistically* is unlikely using only 16S rRNA gen amplicon data. That being said, we have now updated our Discussion section to *hypothesize* on the difference between droplet and plate grown cultures for future investigations to test. The updated text is as follows:

“… Second, parasitism, amensalism, and competition are eliminated between strains since each colony is isolated in its own droplet. […] However, we note that disentangling the interplay of these benefits to determine the exact mechanism which allowed any given organism to grow would likely require mechanistic studies on a case-by-case basis.”

We hope the reviewer finds this satisfactory.

Reviewer #3:[…] The work is technically sound and well-designed. However, a couple of concerns raised in this manuscript should be resolved by the author's clarification for the publication in eLife.– It is unclear if the individual picoliter droplet really contains a single bacterium. The authors claimed the single-bacterium confinement in a droplet, but they never showed an initial entrapment of the bacterium as a single cell level, whereas authors only showed the droplets that contain fully grown bacterial cells. A simple bacterial Live/dead staining can be applied to clearly show the single-cell deposition per droplet.

We thank the reviewer for raising this point. While we have the means to implement the exact experiment the reviewer suggests, we regret that we are not able to deliver any experimental results at this time due the disruption COVID-19 has caused. That said we addressed this issue in silico by adding a new supplementary figure (Figure 2—figure supplement 1) which compares the expected vs. measured percentage of droplets with colonies for the standard Poisson loading process. The agreement between the expected percentage with the measured value, coupled with a uniform cell morphology in the separate droplets, suggests that the live cell loading in droplets is in fact largely single cell. We also would like to point out that single cell loading into droplets is a standard practice in droplet microfluidics, and is a very well established method. Also, in the Materials and methods we noticed an error in the reported live cell density count and have fixed it here. We thank the reviewer for their understanding.

– This study showed a unique anaerobic approach and promising applicability to improve the scalable, droplet-based microfluidics toward the culturomics. However, this approach still heavily relies on the nutrient-rich culture medium that may compromise the rare bacterial cells that require a minimal medium, particular trace elements, or host-derived metabolites. The main reason that the authors could find the higher richness and diversity in the droplet-based isolation compared to the agar plate method is simply that the system more accurately manages the cultured cells in each droplet under the confinement effect and the segregation of bacterial cells to each other, but not because they used a fancy medium. Thus, this method is still limited to claim to culture the unculturable fecal bacteria in vitro. This part should be substantially discussed.

We’ve expanded our discussion on the limitations nutrient-rich culture (Discussion, fourth paragraph). This is part of our “Limitations” paragraph in the Discussion. Additionally, we mention the problem of coloading with obligate symbionts in the second point of our “Limitations”.

“Finally, here we loaded droplets with three different rich media in order to broadly enrich the cultivated community representation across taxa, since the previous surveys found the majority of gut bacteria which grow in defined nutrient limited media also grow in rich media [Tramontano et al., 2018]. However, some bacteria require minimal media with specific carbohydrates, vitamins, or trace elements [Tramontano et al., 2018; Oberhardt et al., 2015]”.

“Second, in this study, we isolated single living bacterial cells into droplets in order to prevent interspecies competition. […] Future studies could stochastically co-encapsulate multiple gut bacteria into droplets (by increasing the loading cell density during droplet generation) and investigate the resulting growth dynamics.”